# Navigating Memory Construction by Global Pseudo-Task Simulation for Continual Learning

**Yejia Liu**[*†]  **Wang Zhu**[*‡]  **Shaolei Ren**[†]

[†]University of California Riverside   [‡]University of Southern California
{yliu807, shaolei}@ucr.edu   wangzhu@usc.edu

## Abstract

Continual learning faces a crucial challenge of catastrophic forgetting. To address this challenge, experience replay (ER) that maintains a tiny subset of samples from previous tasks has been commonly used. Existing ER works usually focus on refining the learning objective for each task with a static memory construction policy. In this paper, we formulate the dynamic memory construction in ER as a combinatorial optimization problem, which aims at directly minimizing the global loss across all experienced tasks. We first apply three tactics to solve the problem in the offline setting as a starting point. To provide an approximate solution to this problem in the online continual learning setting, we further propose the Global Pseudo-task Simulation (GPS), which mimics future catastrophic forgetting of the current task by permutation. Our empirical results and analyses suggest that the GPS consistently improves accuracy across four commonly used vision benchmarks. We have also shown that our GPS can serve as the unified framework for integrating various memory construction policies in existing ER works.

## 1 Introduction

Data in real world is non-stationary and keeps changing. Adapting new knowledge while maintaining the old skills learned from previous data is an idealism for intelligent systems. However, deep artificial neural networks suffer from catastrophic forgetting of old behaviors as the learning of new tasks keeps overwriting the past [30, 32, 33, 41, 34]. To combat this issue, numerous novel algorithms have been designed in continual learning in recent years [37, 13, 28, 8, 20, 2, 27]. Amongst them, the experience replay (ER) methods have been attested as one of the few methods that consistently achieve strong results across different continual learning setups [6, 11, 46, 21]. In the ER, a slightly relieved continual learning setting is considered, where a learner stores a subset of old examples from previous tasks in a fixed-sized memory and jointly trains the memory with the upcoming task.

Most existing ER-based works put rigorous efforts in refining the local learning objective to update the model parameters on one task at a time. In the methods, they stick to a single static memory construction policy, which is prone to failing in the long task sequence. For example, selecting random data examples from experienced tasks for the memory buffer can effect good generalization at the very beginning, but the forgetting rate would soon climb up when the tasks sequence becomes longer, as some class representations are totally squeezed out from the memory [11]. This major drawback has inspired us to explore the optimal memory construction with a dynamic policy.

In this work, we formulate the **memory construction problem** in ER as a combinatorial optimization problem. Unlike previous memory construction methods [1, 3, 15], we explicitly optimize the global objective, *i.e.*, the minimum loss of the final model on all observed tasks, by finding the best memory

---

[*] Author contributed equally

configuration strategy. As a starting point, we approach this problem in an offline setting, where we can go through the task sequence for multiple trials independently. By utilizing three tactics, we reduce the intractable search space to a significantly smaller one. Specifically, for each task, we blend random and class-balanced memories according to a parameter, the *switching point*. Then, we use the binary search to find switching points in $O(T \log |\mathcal{M}|)$, where $T$ is the total number of tasks and $|\mathcal{M}|$ is the size of the memory buffer.

Based on the offline solution, in the online continual learning setup,[2] we propose our method **Global Pseudo-task Simulation (GPS)** as an approximate solution to the problem. The GPS mimics the catastrophic forgetting pattern for the current task by creating future pseudo-tasks, and finds approximate switching points based on the pseudo-task simulation. We examine a few simulation methods and find permutation is the favorable way to synthesize pseudo-tasks.

We conduct experiments on four widely used vision benchmarks. Our results have shown that the GPS achieves higher accuracy compared to baselines, especially when we have a long task sequence. In addition, our empirical analysis verifies that the dynamic memory construction by using GPS is close to the offline solution. Meanwhile, GPS can be easily applied to other ER variants [6, 9] to further improve their performance.

## 2   Preliminary and Notations

**Continual Learning**   In continual learning, the model $f(\theta)$ experiences a stream of data points $(x_i, y_i) \sim P_i$ from a sequence of tasks $t_i$, where $i \in \mathcal{T} = \{1, ..., T\}$, and $P_i$ is an unknown i.i.d. distribution of task $t_i$. Without experience replay, the model $f(\theta)$ is optimized on one task at a time following the task sequence under the tight constraint that the examples from previous tasks cannot be accessed [38]. We denote $\theta_i$ as the parameter of $f(\cdot)$ *after* training task $t_i$, and we refer to the function $g(\cdot)$ that updates $\theta_i$ for each task $t_i$ as the *local updating method*. Once the local updating method is determined, $\theta_T$ can be derived recursively by $\theta_i = g(\theta_{i-1}, P_i)$ from $\theta_0$, which is the initialization point.

After sequentially training $T$ tasks, the objective of continual learning is to achieve the minimum loss across all observed tasks with the final model in the end. We write the summed *global loss* $\mathcal{L}^G$ as in Eqn. (1), where $\theta_T$ is the final parameter after $T$ tasks and $l(\cdot)$ is the cross-entropy loss. For convenience, we also denote the global loss for a single task $i$ as $\mathcal{L}_i^G = \mathbb{E}_{(x_i, y_i) \sim P_i} \ell(y_i, f(x_i; \theta_T))$.

$$\mathcal{L}^G = \sum_{i=1}^{T} \mathbb{E}_{(x_i, y_i) \sim P_i} \ell(y_i, f(x_i; \theta_T)) \tag{1}$$

**Experience Replay**   Previous works [13, 38, 46] have proposed a series of local updating methods to optimize the global objective. Amongst them, one effective way is experience replay (ER) from [11]. Experience replay relieves a bit on the tight constraint in continual learning by adding a fixed-sized memory buffer $\mathcal{M}$ to store a limited subset of seen examples.

We denote the memory *after* training task $t_i$ as $\mathcal{M}_i$. The modified local updating method $g$ treats the memory as another input, and jointly optimizes examples of the current task and examples stored in the memory, with a factor $\lambda$ on the loss of memory examples as shown in Eqn. (2). Thus, $\theta_i$ is iteratively updated by $\theta_i = g(\theta_{i-1}, P_i, \mathcal{M}_{i-1})$.

$$g(\theta, P, \mathcal{M}) = \arg\min_{\theta} \{\mathcal{L}_t(\theta, P) + \lambda \mathcal{L}_t(\theta, \mathcal{M})\} \tag{2}$$

$$\mathcal{L}_t(\theta, P) = \mathbb{E}_{(x, y) \sim P} \ell(y, f(x; \theta)) \tag{3}$$

Both empirical results [11, 6] and theoretical analysis [21] have suggested that the local updating method $g$ of experience replay is effective on reducing the global loss $\mathcal{L}^G$. If not specially specified, we refer to local updating method $g$ as Eqn. (2) in the following text.

## 3   Problem Formulation: Dynamic Memory Construction

Most previous works based on ER regard the global loss $\mathcal{L}^G$ as a function of $g$ with a static memory construction strategy for $\mathcal{M}$. They utilize various techniques like regularization [6, 9] or memory

---

[2]We use *online* as opposed to the multi-trial offline setup, while we still use the multi-pass training here.

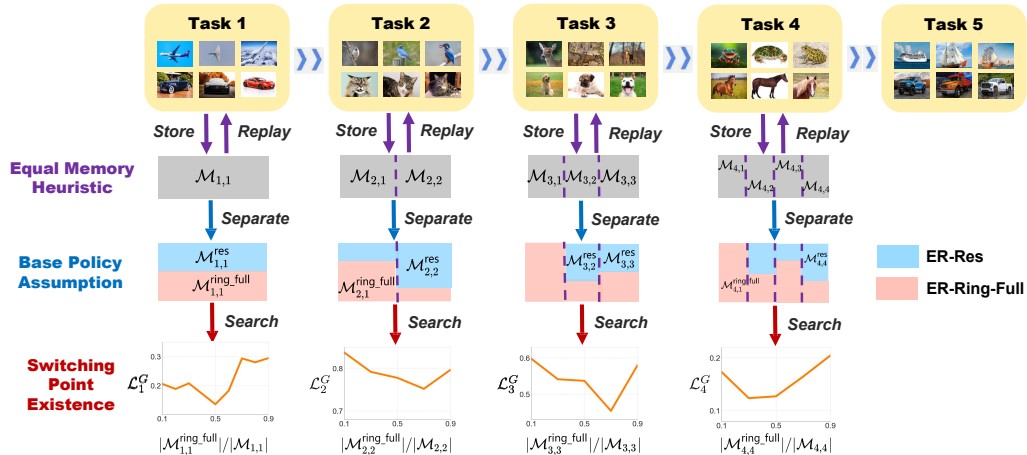

Figure 1: Using three tactics on the S-CIFAR-10 dataset to solve the dynamic memory construction problem in the offline setting. The equal memory heuristic and the base policy assumption reduce the search space. The switching point existence enables the binary search. For each task, we split the memories into a random part (ER-Res) and a class-balanced part (ER-Ring-Full) according to the switching point.

sampling [1] to further refine the local updating method $g$. Unlike them, we view the global loss $\mathcal{L}^G$ as a function of the memory $\mathcal{M}$, and explicitly minimize $\mathcal{L}^G$ by optimizing the memory *without* modifying the local updating methods in Eqn. (2).

Following the setup of ER, we propose dynamic memory construction, which aims at finding the best memory construction $\mathcal{M}$ to optimize the global objective $\mathcal{L}^G$, as defined in Eqn. (1). This combinatorial optimization problem is expressed by the Eqn. (4), where we minimize the global objective by considering the memory $\mathcal{M}_i$ after task $t_i$ as variables.

$$\min_{\{\mathcal{M}_i\}_{i\in\mathcal{T}}} \mathcal{L}^G(\{\mathcal{M}_i\}_{i\in\mathcal{T}}) \tag{4}$$

We denote the part in $\mathcal{M}_i$ storing the examples of task $t_j$ as $\mathcal{M}_{i,j}$. There are four intrinsic constraints for the optimization problem. In all constraints, $i, j, i' \in \mathcal{T}$.

- $\mathcal{M}_i$ is the union of memories $\{\mathcal{M}_{i,j}\}_{j\in\mathcal{T}}$. The collection $\{\mathcal{M}_{i,j}\}_{j\in\mathcal{T}}$ is mutually disjoint.

- No future examples in the memory, *i.e.*, $\mathcal{M}_{i,j} = \varnothing$ when $i < j$.

- The size of memory is the same across the training procedure from $t_1$ to $t_T$, denoted as $|\mathcal{M}|$.

- As we cannot access previous examples that are not stored in the memory, once the set for task $t_j$ in the memory is constructed after training task $t_j$, its size will not increase. Thus, the memory set for task $j$ in $\mathcal{M}_i$ is always a subset of that in $\mathcal{M}_{i'}$ given $j \leq i' < i$.

## 4 An Offline Solution: Reduce the Search Space

In order to reduce the intractable search space of $\mathcal{M}_\mathcal{T}$ quantitatively, we first consider the dynamic memory construction problem in an offline setup as the starting point for analyzing the realistic online setting. In the offline setup, we can go through the task sequence for multiple trials independently, but we are still strictly not allowed to access previous examples that are not in $\mathcal{M}$ in each trial. We take three tactics to reduce the search space, referred to as the *equal memory heuristic*, the *base policy assumption* and the *switching point existence*. Fig. 1 illustrates how we apply three tactics to the S-CIFAR-10 dataset. Based on these tactics, we provide an approximate solution to the problem.

## 4.1 Equal Memory Buffer for Each Task

We first take the equal memory heuristic from [11], which has shown that an equal size of memory buffer for each observed task is effective to avoid catastrophic forgetting.[3] Based on this tactic, we add $|\mathcal{M}_{i,j}| = |\mathcal{M}|/i$, where $i \geq j$, to the previous constraints list. Given a fixed size of each $\mathcal{M}_{i,j}$, instead of optimizing $\mathcal{M}_i$ from previous observed tasks jointly, we can independently optimize each $\mathcal{M}_{i,j}$ and construct $\mathcal{M}_i$ from $\mathcal{M}_{i,j}$. To this end, we replace the optimization objective Eqn. (4) by a simplified version Eqn. (5), taking $\mathcal{M}_{i,j}$ as variables.

$$\min_{\{\mathcal{M}_{i,j}\}_{i,j \in \mathcal{T}}} \mathcal{L}^G(\{\mathcal{M}_{i,j}\}_{i,j \in \mathcal{T}}) \tag{5}$$

However, given the equal memory heuristic, the search space in Eqn. (5) is still very large. For each task $t_j$, suppose the total number of examples for task $t_j$ is $n_j$ and $n_j > |\mathcal{M}|$, $\mathcal{M}_{i,j}$ has more than $\binom{|\mathcal{M}|}{|\mathcal{M}|/i}$ different constructions.

## 4.2 Mix of Base Policies

To further reduce the search space in Eqn. (5), we introduce two base policies, ER-Res and ER-Ring-Full [11]. For a given task $t_i$, ER-Res builds a memory by random samples, while ER-Ring-Full builds a memory by selecting the same number of samples from each class.

We take the base policy assumption, where we deem that the memories having the same size and taking the same base policies are the same. It implies that even though two memory buffers contain different data examples, they are still identified as the same as long as they meet these two rules.

This assumption ignores randomness of data selection within the same policy. It is backed up by analysis in DER [6], which shows the standard deviation of accuracy on different benchmarking datasets is quite small ($\sim$0.5) when using the same policy with a relatively large $|\mathcal{M}|$. Based on the assumption, we separate $\mathcal{M}_{i,j}$ into two disjoint parts and take the mixed policy, where $\mathcal{M}_{i,j}^{\text{res}}$ and $\mathcal{M}_{i,j}^{\text{ring-full}}$ represent two parts in $\mathcal{M}_{i,j}$, namely the former taking the ER-Res and the latter taking ER-Ring-Full policies, respectively. For completeness, we extend the subset constraint to $\mathcal{M}_{i,j}^{\text{res}} \subseteq \mathcal{M}_{i',j}^{\text{res}}$, $\mathcal{M}_{i,j}^{\text{ring-full}} \subseteq \mathcal{M}_{i',j}^{\text{ring-full}}$, where $j \leq i' < i$.

Previous studies of ER [11] have shown the randomness (ER-Res) is crucial in a large memory, while guaranteeing the equal representation of each class (ER-Ring-Full) is crucial under a tiny memory. As the number of tasks grows, the size of memory for each task $t_i$ becomes smaller. Under a mixed policy, it is natural to first shrink the ER-Res part of the memory and then shrink the ER-Ring-Full part of the memory. By doing so, given $\mathcal{M}_{j,j}^{\text{res}}$ and $\mathcal{M}_{j,j}^{\text{ring-full}}$, we can determine $\mathcal{M}_{i,j}^{\text{res}}$ and $\mathcal{M}_{i,j}^{\text{ring-full}}$ accordingly for each $i > j$. We denote the size of $\mathcal{M}_{j,j}^{\text{ring-full}}$ as $a_j$ and derive the size of $\mathcal{M}_{j,j}^{\text{res}}$ as $(|\mathcal{M}|/j) - a_j$. Substituting the collection of sets $\{\mathcal{M}_{i,j}\}_{i,j \in \mathcal{T}}$ by an integer variable $a_j$, we therefore further simplify the optimization objective, as in Eqn. (6).

$$\min_{\{a_j\}_{j \in \mathcal{T}}} \mathcal{L}^G(\{a_j\}_{j \in \mathcal{T}}) \tag{6}$$

where each $a_j$ has $|\mathcal{M}|/j$ different choices, significantly smaller than the search space for $\mathcal{M}_{i,j}$ in Eqn. (5). Notice that the memory construction for the last task $t_T$ is not required. Thus, the total search space equals to $|\mathcal{M}|^{T-1}/(T-1)!$, which is still large.

## 4.3 Existence of a Switching Point

For the memory allocation after each task $t_j$, there exists a gold point $a_j = s_j$ that assigns exactly the *required* ring-full memory to keep the best balance of $\mathcal{M}_{i,j}^{\text{ring-full}}$ and $\mathcal{M}_{i,j}^{\text{res}}$ for the following task $t_i$. If the ring-full memory size is not large enough, we lose the power of forcing an equal representation of classes as the task sequence grows. Conversely, if the ring-full memory is more than enough, we sacrifice the randomness when the task number is still small. To this end, we assume $s_j$ is a unique switching point, *i.e.*, there exists a switching point $s_j$ for each $a_j$, which satisfies the monotonicity

$$\mathcal{L}^G(\{a_j'\} \cup \{a_i\}_{i \in \mathcal{T}/\{j\}}) > \mathcal{L}^G(\{a_j\} \cup \{a_i\}_{i \in \mathcal{T}/\{j\}}), \quad a_j' < a_j < s_j \text{ or } s_j < a_j < a_j' \tag{7}$$

---

[3]For simiplicity of the formulation, we assume each task has the same number of classes in $\mathcal{M}$.

Notice that the global loss of task $t_j$ should depend closely on what data we store and replay for task $t_j$, but very loosely on what data we store and replay for other tasks. Following this intuition, though it is hard to verify a specific point satisfies the condition of Eqn. (7) for all combinations of $\{a_i\}_{i \in \mathcal{T}/\{j\}}$, we can simplify the switching point condition to

$$\mathcal{L}_j^G(a_j') > \mathcal{L}_j^G(a_j), \qquad\qquad a_j' < a_j < s_j \ \text{ or } \ s_j < a_j < a_j' \qquad (8)$$

Given the switching point existence, though we still have the same search space, instead of a linear search, we can apply a binary search algorithm to reduce the time complexity to $O(T \log |\mathcal{M}|)$. During the binary search, if we search by comparing against the exact left and right integer of the point, i.e., comparing the loss computed by $a_j + 1, a_j - 1$ and $a_j$, the large variance of sampling may cause the algorithm to exit unexpectedly. To increase the robustness of the algorithm, we take a search stride $\epsilon$, where we compare the loss computed by $a_j + \epsilon, a_j - \epsilon$ and $a_j$. The stride is determined by the benchmark and the size $|\mathcal{M}_j|$. The detailed binary search algorithm for $s_j$ can be found in our Appendix A.

This assumption is valid for over 85% of the cases from the observations in Fig. 1 (for the S-CIFAR-10 dataset) and Appendix B (for other benchmarks). For each $a_j$ (i.e., $|\mathcal{M}_{j,j}^{\text{ring-full}}|$) in the figures, the global loss $\mathcal{L}^G$ initially decreases monotonically and then increases monotonically as $a_j$ grows. Rarely but possible, due to the variance of sampling, we cannot find the switching point $s_j$. In such cases, we will choose $a_j$ with the minimum loss amongst the values we searched.

# 5 Global Pseudo-task Simulation (GPS)

The offline solution of dynamic memory construction requires going through the task sequence for $O(T \log |\mathcal{M}|)$ times to find $\{s_j\}_{j \in \mathcal{T}}$, which violates the online continual learning setup. To circumvent this issue, we propose Global Pseudo-task Simulation (GPS), which provides an approximate solution $\{\tilde{s}_j\}_{j \in \mathcal{T}}$ to the problem by simulating the future training process under the online setup. Specifically, we simulate the local updating process Eqn. (2) by creating *pseudo-future tasks*.

## 5.1 Objective Function for Simulation

Perfectly simulating the future is a mission impossible in continual learning [21]. As we have no information about the future tasks, the distribution of the pseudo-future tasks we create could be quite different from the distribution of the real future ones. To find each approximated switching point $\tilde{s}_j$ more precisely, we intend to use more real tasks and less pseudo-future tasks as possible. As we are required to allocate examples of task $t_j$ to a non-empty set $\mathcal{M}_{j,j}$ right after training the task $t_j$, we solve $\tilde{s}_j$ by a simulation process from $\theta_j$ to $\theta_T$, without future modifications. In theory, we could overwrite the previous switching point $s_j$ by a more accurate simulation after task $t_i, i > j$. However, one risk of overwriting the $s_j$ is that we may not have enough examples of task $t_j$ in the memory to support the reallocation brought by the new switching point, since more examples of task $t_j$ cannot be accessed anymore except the ones stored in $\mathcal{M}_{j,j}$. Another drawback comes from the drastically increased simulation complexity if we take the overwriting mechanism.

To this end, we modify the offline objective function Eqn. (6) to the online simulation objective Eqn. (9), where $\tilde{\theta}_{j:i}$ is the simulated $\theta_i$ from the real $\theta_j$, for $i > j$.

$$\tilde{s}_j = \arg\min_{a_j} \mathbb{E}_{(x_j, y_j) \sim P_j} \ell(y_j, f(x_j; \tilde{\theta}_{j:T})) \qquad (9)$$

Note that $\tilde{\theta}_{j:T}$ is derived recursively from $\tilde{\theta}_{j:i} = g(\tilde{\theta}_{j:(i-1)}, \tilde{P}_i, \tilde{\mathcal{M}_{i-1}})$, initialized with $\tilde{\theta}_{j:j} = \theta_j$. $\tilde{P}_i$ is the task distribution of the synthesized pseudo-tasks $\tilde{t}_i$, and $\tilde{\mathcal{M}}_i$ is the simulated pseudo-memory after training the pseudo-task $\tilde{t}_i$. In the online setup, we cannot access the previous tasks as well as the future tasks. Thus, we will not evaluate the summed global objective $\mathcal{L}^G$. Following the offline Eqn. (8), we evaluate on $\mathcal{L}_j^G$, the global objective of task $t_j$.

## 5.2 Synthesizing Pseudo-tasks

The goal of our pseudo-task simulation is to find $s_j$ precisely, i.e., $\tilde{s}_j \approx s_j$. Concretely, instead of accurately simulating the future training process, we use synthesized pseudo-tasks to *mimic the*

*forgetting patterns of task $t_j$ caused by the future tasks.* To achieve this aim, we believe if the real task sequence holds certain properties, it is essential for pseudo-tasks to hold the same set of properties to mimic the same forgetting pattern.

We find most of the existing widely used vision CL benchmarks [6, 25, 43] hold two properties: 1) similar learning difficulty of individual tasks; 2) limited zero-shot transfer ability. The experimental validation of these properties is in Appendix C. To induce the same level of catastrophic forgetting, we expect a pseudo-task $\tilde{t}_j$ to have similar difficulty as its real counterpart $t_j$ in the future, measured by the accuracy in an identical end-to-end training setup [34]. To avoid the similarity between tasks to lead to little forgetting after training on pseudo-tasks, pseudo-tasks should also have low zero-shot accuracy with the model trained on the current task. If pseudo-tasks can achieve high zero-shot accuracy with the model trained on the current task, the similarity between tasks could result in very little forgetting after training on the pseudo-tasks, hindering our purpose of mimicking the catastrophic forgetting.

Based on these two properties, we synthesize pseudo-future tasks from the task $t_j$ by applying different permuting seeds to its input $x_j$ to create a series of future tasks $\{\tilde{t}_{j+1}, ..., \tilde{t}_T\}$. We achieve this by permuting the pixels of each image on image datasets, i.e., a fresh permutation would be generated and applied to all images within a synthesized task [54].

Besides permutation, we have also considered two other synthesizing techniques that lack a certain property we discussed for comparison. One is rotation, where we rotate the image inputs of the task $t_j$ gradually by 15 degrees to create pseudo-future tasks [6]. The other is blurring, where we apply the Gaussian blurring into the image using a $5 \times 5$ filter with growing standard deviation. Rotation creates pseudo-tasks with similar difficulties as the real tasks, but the zero-shot transfer ability is far too good. Blurring creates pseudo-tasks with limited zero-shot transfer ability, yet the task difficulty is increasing along the pseudo-task sequence.

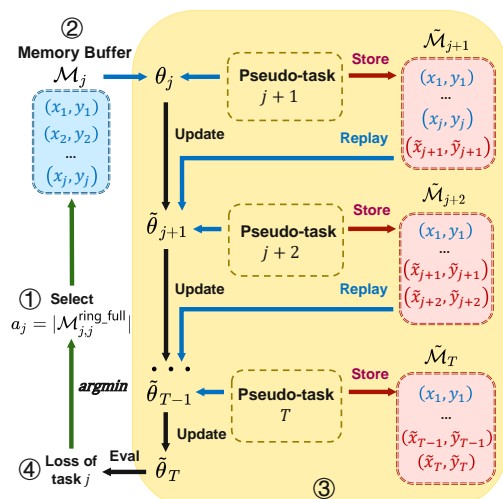

Figure 2: The Global Pseudo-task Simulation process to solve $a_j$ after training task $t_j$.

## 5.3 Construct Pseudo-memories

Fig. 2 visualizes the process of pseudo-task training and pseudo-memory construction. During the simulation process after task $t_j$, we try different $a_j$ with a binary search and pick the best one as $\tilde{s}_j$ based on the online objective Eqn. (9).

We start from $\tilde{\mathcal{M}}_j$, which is the same as the real $\mathcal{M}_j$. The construction for $\tilde{\mathcal{M}}_i$ $(i > j)$ is different for real tasks and pseudo-tasks. For real tasks $t_{j'}, j' \in \{1, ..., j\}$, $\tilde{\mathcal{M}}_{i,j'}$ is the real $\mathcal{M}_{i,j'}$ determined for a given $a_{j'}$, as we discussed in § 4.2. Note that the approximated switching points $\{\tilde{s}_{j'}\}_{j' \in \{1, ..., j-1\}}$ for the previous $j - 1$ tasks are solved before $\tilde{s}_j$. For pseudo-tasks $\tilde{t}_{j'}, j' \in \{j+1, ..., i\}$, $\tilde{\mathcal{M}}_{i,j'}$ is constructed by randomly selecting $|\mathcal{M}|/i$ pseudo-data from $\tilde{t}_{j'}$.

## 5.4 Other Implementation Details

For the efficiency of simulation, the number of examples of the synthesized pseudo-tasks we use in the simulation is equal to $|\mathcal{M}|$. As the simulation quickly converges on a small number of examples, we train fewer epochs for each task compared with the real training process. Besides, when the number of tasks $T$ are large or unknown, we extend GPS by simulating a fixed-sized sliding window of future tasks for the sequence, *e.g.*, simulating 10 pseudo-future tasks for each task. Also, during the local update of each task $t_i$, for a smooth transition, we allow the current training task $t_i$ to take

Table 1: Accuracy of GPS using different simulation techniques and baselines on four vision benchmarks. ER-Oracle shows the performance of the offline solution as described in § 4. Reported numbers are all averaged over 5 runs.

| Method $|\mathcal{M}|$ | Simulation | P-MNIST 1000 | S-CIFAR-10 200 | S-CIFAR-100 2000 | TinyImageNet 2000 |
|---|---|---|---|---|---|
| ER-Res | - | $86.55_{\pm 0.48}$ | $92.01_{\pm 0.80}$ | $81.38_{\pm 0.51}$ | $57.50_{\pm 0.54}$ |
| ER-Ring-Full | - | $84.33_{\pm 0.65}$ | $91.53_{\pm 0.56}$ | $81.16_{\pm 0.65}$ | $54.73_{\pm 0.32}$ |
| ER-Hybrid | - | $86.84_{\pm 0.35}$ | $92.06_{\pm 0.89}$ | $81.47_{\pm 0.23}$ | $57.97_{\pm 0.44}$ |
| **GPS** | **Permutation** | $\mathbf{87.93}_{\pm \mathbf{0.21}}$ | $\mathbf{92.77}_{\pm \mathbf{0.39}}$ | $\mathbf{82.46}_{\pm \mathbf{0.33}}$ | $\mathbf{59.26}_{\pm \mathbf{0.31}}$ |
| | Rotation | $85.38_{\pm 0.20}$ | $91.61_{\pm 0.49}$ | $81.50_{\pm 0.42}$ | $57.45_{\pm 0.33}$ |
| | Blurring | $86.03_{\pm 0.31}$ | $91.96_{\pm 0.38}$ | $81.49_{\pm 0.46}$ | $56.85_{\pm 0.27}$ |
| ER-Oracle | Offline | $88.26_{\pm 0.15}$ | $93.09_{\pm 0.35}$ | $82.88_{\pm 0.31}$ | $60.56_{\pm 0.23}$ |

up to $|\mathcal{M}|/i$ the size of the memory without changing $\mathcal{M}_{i,j}$ for any previous task $t_j$.[4] We also put the detailed algorithm for GPS in Appendix A.

# 6 Experiments

## 6.1 Experimental Setup

**Datasets** We carry out evaluations on four widely used vision benchmarks in continual learning, P-MNIST, S-CIFAR-10, S-CIFAR-100 and TinyImageNet [43, 6, 25]. P-MNIST was proposed in [18]. It contains 10 tasks where the first task is the MNIST dataset [23] while the later ones are constructed by permuting each image in MNIST with an unique permutation seed. The S-CIFAR-10 is constructed by splitting CIFAR-10 [17] into 5 sequential tasks where each task contain 2 classes and 12,000 images [22, 6]. Similarly, we split CIFAR-100 [17] into 10 tasks where each one contains 10 classes and 6,000 images to construct S-CIFAR-100. The TinyImagenet [48] is a subset of ImageNet [14] with 200 classes. We split it into 10 consecutive tasks with 20 classes per task.

**Architectures** For P-MNIST, we apply a fully connected network with two hidden layers. Each comprises 100 ReLU units. For S-CIFAR-10, S-CIFAR-100 and TinyImageNet, we use Resnet18 following [6] and [13].

**Baselines** The baselines we used in experiments include ER-Res, ER-Ring-Full and ER-Hybrid [11]. ER-Hybrid is a mix of ER-Res and ER-Ring-Full, where the memory construction strategy would switch from the former to the latter once observing only one sample of some class is left in $\mathcal{M}$. We also compare to non-ER methods: online EWC (oEWC) [44], iCaRL [37], A-GEM [10] and GSS [3].

**Training Details:** Our training all use stochastic gradient descent (SGD) with a learning rate of 0.1. We use $\lambda = 1$ in the local updating method Eqn. (2). For P-MNIST, we train 5 epochs for each task while increasing the number of epochs to 50 for S-CIFAR-10, 100 for both S-CIFAR-100 and TinyImageNet regarding their data complexity, as done by works [6, 43]. For P-MNIST and S-CIFAR-10, we set the batch size as 10. For S-CIAFR-100 and TinyImageNet, batch size is set to 50. Following the implementation of ER [11], we set the same batch size for training the current task and the memory. Note the permutation seeds we use for simulation in P-MNIST are different from the ones used in P-MNIST itself to avoid peeping into test datasets. More training details and hyperparameter values can be found in the Appendix F.

## 6.2 Main Results

In Table 1, we compare GPS to ER baselines and the offline oracle on the four vision benchmarks. We also compare GPS using different simulation techniques.

---

[4]We release our code at `https://github.com/liuyejia/gps_cl`

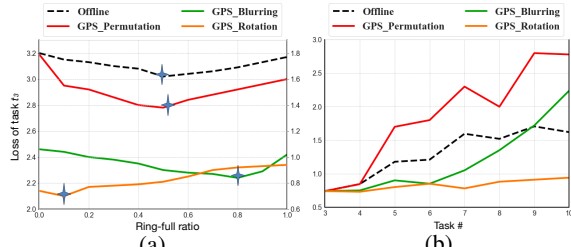

| Method | $T = 20$ | $T = 40$ |
|---|---|---|
| ER-Res | $71.25_{\pm 1.07}$ | $47.33_{\pm 2.75}$ |
| ER-Ring-Full | $71.92_{\pm 1.47}$ | $49.26_{\pm 2.66}$ |
| ER-Hybrid | $72.27_{\pm 0.88}$ | $49.63_{\pm 2.61}$ |
| **GPS** | $\mathbf{74.63_{\pm 0.20}}$ | $\mathbf{53.57_{\pm 0.63}}$ |
| ER-Oracle | $75.21_{\pm 0.17}$ | $54.44_{\pm 0.51}$ |

(a)       (b)       (c)

Figure 3: (a). Global loss of task $t_3$ from the S-CIFAR-100 benchmark w.r.t. different configurations of $\mathcal{M}_{3,3}$ (or $\tilde{\mathcal{M}}_{3,3}$). The Offline and Rotation curves use the right y-axis, while the Permutation and Blurring curves use the left y-axis. The blue stars mark the switching points. (b). The loss of task $t_3$ after training future tasks. (c). Accuracy of GPS and baselines on long sequence P-MNIST.

**GPS using permutation performs better**   GPS using permutation shows improved accuracy compared to ER baselines. Its performance is close to the performance of the offline oracle solution, which implies it is a good approximation. We can also see that GPS using permutation performs better than using rotation and blurring, in terms of both accuracy and stability. The results support our hypotheses on the properties that synthesized tasks should bear, *i.e.*, similar level difficulties to previous tasks and limited zero-shot transfer ability.

**Dynamic memory construction provides stability**   Besides, we notice that the offline ER oracle, and GPS using different simulation techniques, are more stable than ER baselines as they achieve lower standard deviation on most of the evaluated benchmarks. We attribute this to the reduced interference from random seeds as we take the global objective into explicit consideration.

### 6.3 Analysis of Simulation Methods

To further understand why permutation works better than other simulation techniques, we analyze the task $t_3$ of the S-CIFAR-100 benchmark. To make fair comparisons with the offline oracle, we train the pseudo-tasks with the same amount of training data and training epochs here. We plot the curve of global loss w.r.t. the ratio of ER-Ring-Full in the memory $\mathcal{M}_{3,3}$ (or pseudo memory $\tilde{\mathcal{M}}_{3,3}$), and the forgetting curve of task $t_3$ after training future tasks (or future pseudo-tasks), as in Fig. 3.

We can see that the rotation induces less forgetting (lower loss) than the real tasks (offline cases) along the sequence, as shown in Fig. 3(b), as rotation creates tasks sequences that bear good zero-shot transfer from the previous tasks. The pseudo-task sequence created by rotation is therefore too easy such that the switching point of the curve is close to 0, as shown in the Fig. 3(a), which implies it is similar to an ER-Res static policy. As for the blurring, the forgetting curve first climbs slowly, as it allows some zero-shot transfer from the previous tasks. And then, the loss increases dramatically since the task difficulty goes up. Its pattern of forgetting is quite different from the real tasks (offline case) as shown in the Fig. 3(b).

We also observe that the GPS using permutation has the closest switching point to the offline compared to the rest. Though permutation creates pseudo-task sequence that result in higher losses than the real task sequence, the pattern of forgetting is similar. We compute the L1-norm of the offline sampling ratios *vs*. permuting-simulated sampling ratios, which are 0.9, 0.7, 1.2, 1.2 for P-MNIST, S-CIFAR-10, S-CIFAR-100 and TinyImageNet, respectively. The method selects sampling ratios for other benchmarks as good as that for the P-MNIST, which implies the similarity of pseudo-tasks is not the major factor for the enhanced results.[5].

### 6.4 Long Task Sequence

We extend the 10-task P-MNIST benchmark to 20 and 40 tasks to create longer task sequences, and make the same comparison on GPS, ER-Oracle and three ER baselines in Fig. 3(c).

---

[5]In the following text, we refer to "GPS using permutation" as GPS.

Table 2: (a). Time cost (in minutes) of training *vs*. simulation for P-MNIST and S-CIFAR-10. (b). Accuracy of GPS when incorporating existing ER variants, DER++ [6] and HAL[9], comparing to other methods. For these two ER variants, we use a memory size of 1000 on P-MNIST and a memory size of 2000 on the TinyImageNet.

| Short Seq | # Tasks | Training | Simulation |
|---|---|---|---|
| **P-MNIST** | 10 | 26.32 | 2.30 |
| **S-CIFAR-10** | 5 | 545.56 | 10.02 |
| **S-CIFAR-100** | 10 | 1187.93 | 137.63 |
| **TinyImageNet** | 10 | 2418.20 | 209.98 |
| Long Seq | # Tasks | Training | Simulation |
| **P-MNIST** | 20 | 55.29 | 6.21 |
| **P-MNIST** | 40 | 108.17 | 13.37 |

(a)

| | P-MNIST | TinyImageNet |
|---|---|---|
| oEWC | $69.21_{\pm 2.92}$ | $20.81_{\pm 0.95}$ |
| iCaRL | - | $38.77_{\pm 3.68}$ |
| GSS | $86.34_{\pm 4.28}$ | - |
| A-GEM | $77.36_{\pm 1.28}$ | $25.30_{\pm 0.87}$ |
| OGD | $81.52_{\pm 2.21}$ | - |
| HAL | $87.69_{\pm 0.34}$ | - |
| **GPS+HAL** | $\mathbf{88.23_{\pm 0.03}}$ | - |
| DER++ | $91.14_{\pm 0.22}$ | $60.67_{\pm 1.08}$ |
| **GPS+DER++** | $\mathbf{91.64_{\pm 0.16}}$ | $\mathbf{61.01_{\pm 0.98}}$ |

(b)

**Longer task sequence requires more careful memory construction** The offline solution, *i.e.*, ER-Oracle, outperforms the baseline policies by $\sim 5\%$ accuracy when $T = 40$. We attribute the performance gain to the global objective oriented memory construction. Clearly, for our global objective oriented memory construction, longer task sequence means larger search space of memory construction. Notice for ER baselines in Fig. 3(c), the standard deviations are larger than those on shorter task sequences, which implies that the larger space of memory construction choices makes a larger performance gap between the worst and the best memory construction.

We also see that GPS solves the problem significantly better than the baselines. Interestingly, GPS still achieves close performance to the offline solution when the task sequence is long.

## 6.5 Simulation Time Cost

We show how many extra time, *i.e.*, pseudo-task generation and pseudo-task training time, GPS adds to the standard ER training. We compute the time cost of the standard ER training *vs*. the extra time from GPS in Table 2(a).

**GPS is time efficient** We take the asynchronous simulation, which applies a binary search to determine $\tilde{s}_j$ sequentially in $O(T \log |\mathcal{M}|)$ sweeps. Suppose simulating the training of each pseudo-task takes a unit time, each sweep is in $O(T)$. Then, the total simulation process is in $O(T^2 \log |\mathcal{M}|)$, which is dependent on the memory buffer size and the number of simulation training epochs. To ensure efficiency, we train pseudo-tasks with fewer epochs. We disclose those values for each dataset in Appendix F. When the task sequence is short, from Table 2(a), we can see the simulation time is over ten times less than the standard training time. When the task sequence is long, we can see the simulation cost grows linearly w.r.t. the number of tasks, as we restrict the search window from $T - j$ to 10 to prevent the simulation cost increasing quadratically.

## 6.6 Exploration of Other Local Updating Methods

We show the performance of adopting other local updating methods from the existing ER variants besides Eqn. (2), together with GPS. The exemplar base policies we take are from the DER++ and HAL. DER++ leverages knowledge distillation in the episodic memory construction. More specifically, the DER++ stores both the network output logits of examples and their ground truth labels in the memory. HAL selects pivotal learned data points besides random samples to store in the memory. We change the local updating method from Eqn. (2) to the local updating method used in DER++ or HAL respectively to transplant them into GPS, without any other modifications to the algorithm.

**GPS integrates well with advanced ER variants** From Table 2(b), we can see the performance of DER++ and HAL has been further improved by taking the optimized memory construction for $\mathcal{M}$

by GPS, which implies the ability of GPS as a framework to incorporate advanced memory-based continual learning methods. Besides, as the state-of-the-art method DER++ outperforms other non-ER methods, GPS+DER++ naturally outperforms them.

# 7 Related Works

**Continual Learning**   Enabling an intelligent agent to learn progressively and adaptively without forgetting old knowledge is a long-standing objective in AI [49, 39]. To combat the catastrophic forgetting problem [30, 18], a few methods have been proposed in continual learning [12, 2, 13]. They usually can be categorized into three classes. One is the regularization-based methods, which introduce an additional regularization term in the loss function to consolidate old behaviors when learning new tasks [20, 44, 53, 8, 19, 36, 15]. One is the replay methods, which store a tiny amount of old examples in a size-bounded memory or condense previous knowledge in a generative model to generate pseudo samples [26, 46, 1, 9, 6, 11, 40, 1, 24, 47, 51, 7, 35]. The data stored in the buffer would be revisited and trained together with each current training task. The third is the parameter isolation methods, which usually allocate additional neural resources for new knowledge without constraints on the model size [42, 29, 45, 52]. The memory cost of these methods would therefore scale with the number of tasks.

**The ER Family**   Experience replay falls into the replay method category, which takes a fixed-sized buffer to store old examples. Existing experience replay variants have adopted the same setup as ER, and focus on refining the local updating method, *i.e.* how to optimally update model parameters by joint training of current task and examples in memory. The advanced techniques used to improve local updating step include additional regularization [9], knowledge distillation [5, 6] and selective memory sampling [1, 35]. Distinct to vanilla ER or its variants, which implicitly minimize the global loss of the final model parameters by designing a powerful local updating method, our work focus on directly optimizing the global objective function by creating pseudo-tasks to mimic the catastrophic forgetting for the current task.

# 8 Conclusion

In this paper, we propose the dynamic memory construction optimization problem for continual learning under the experience replay setup. The problem aims at finding the best memory construction strategy to optimize the global objective function in continual learning. We simplify the problem to a small space by taking three tactics, and find a solution in time complexity $O(T \log |\mathcal{M}|)$ in an offline setup. We then officially introduce our Global Pseudo-task Simulation (GPS), which provides an approximate solution to the simplified problem under the realistic online setup by creating pseudo-tasks to mimic the future catastrophic forgetting pattern for the current task. Our empirical results have shown our approach outperforms baselines and can improve the accuracy of existing ER variants [9, 6].

**Future Studies**   We focus on the task- and domain-incremental in this work. Some future improvements could be extending Global Pseudo-task Simulation to the class-incremental (class-IL) setup. There are two potential challenges under this setup. 1) the task identity is known. This could be achieved by triggering the simulation after experiencing a certain amount of data points instead of a task. 2) though the ER-Res and ER-Ring-Full mixed policies are good enough under task and domain IL, we might involve a complex mixture policies to find the offline oracle under class-IL, where new assumptions are required.

Further, we hope our work could inspire the community to design new datasets closer to the real-world setting. For example, when task sequences have specific zero-shot transfer patterns. In such cases, we might first infer the zero-shot transfer pattern from a few data points and then inject the bias into the simulation process.

# Acknowledgements

We would like to express our sincere gratitude to Sébastien M. R. Arnold, Robin Jia and Jesse Thomason from USC for their valuable suggestions on writing.

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
