# Appendix

In this appendix, we provide details skipped in the main text. The content is organized as follows:

- Section A. Detailed algorithms of GPS, including binary search and simulation process. (*c.f.* §5.4 of the main text)
- Section B. Validation of the switching point existence. (*c.f.* §4.3 of the main text)
- Section C. Validation on the properties of the simulation methods. (*c.f.* §5.2 of the main text)
- Section D. Additional experimental results and ablation studies. (*c.f.* §6.2, §6.5 and §6.6 of the main text)
- Section E. Exploration of new base policies based on curriculum learning, and their performance with GPS.
- Section F. Experimental details and hyperparameters. (*c.f.* §6.1 of the main text)
- Section G. Algorithms of ER-Res and ER-Ring-Full.

## A  Detailed Algorithms

### A.1  Simulation Process

The pseudocode of our simulation process (Fig. 2 in the main text) is listed in Algorithm 1. We use the notation $P_{1:i}$ to represent the task distributions from $P_1$ to $P_i$. Likewise, we use the notation $s_{1:i}$ to represent the switching point from $s_1$ to $s_i$. The memory construction function `BuildM` takes as arguments the previous memory, the current task distribution, and an optional current switching point. If the the switching point is provided, this function internally construct the memory with the given switching point as described in §4; otherwise, it utilizes the described pseudo-memory construction methods as described in §5.

---

**Algorithm 1** GlobalSim

**Input:** Tested point $a_i$; Pseudo-task distributions $\tilde{P}_{(i+1):T}$; Local updating method $g$; Current model parameters $\theta_i$ and current memory $\mathcal{M}_{i-1}$.
Initialize $\tilde{\theta}_{i:i} \leftarrow \theta_i$
Initialize memory $\tilde{\mathcal{M}}_i \leftarrow \text{BuildM}(\mathcal{M}_{i-1}, a_i)$
$j \leftarrow i + 1$
**while** $j \leq T$ **do**
  Local update: $\tilde{\theta}_{i:j} \leftarrow g(\tilde{\theta}_{i:(j-1)}, \tilde{P}_j, \tilde{\mathcal{M}}_{j-1})$
  Build memory: $\tilde{\mathcal{M}}_j \leftarrow \text{BuildM}(\tilde{\mathcal{M}}_{j-1})$
  $j \leftarrow j + 1$
**end while**
Compute the loss: $l \leftarrow \mathbb{E}_{(x_j, y_j) \sim P_j} \ell(y_j, f(x_j; \tilde{\theta}_{j:T}))$
Compute the accuracy $acc_j$ of task $t_j$ with model parameter $\tilde{\theta}_{j:T}$
**return** Loss: $l$, Accuracy: $acc_j$

---

### A.2  Binary Search

In the global binary search as listed in Algorithm 2. To increase the robustness of the algorithm, we take a minimum search stride $min\_\epsilon = 20, 10, 40, 40$ and a maximum search stride $max\_\epsilon = 100, 20, 200, 200$ for four benchmarks P-MNIST, S-CIFAR-10, S-CIFAR-100, TingImageNet, respectively. Also, as we evaluate the continual learning algorithms on the mean accuracy over $T$ tasks, we apply the binary search to find switching point with the highest accuracy but not the lowest loss. Though lower loss usually implies higher accuracy, directly searching based on accuracy gives us slightly better performance.

### A.3  GPS Algorithm

Based on the simulation process and binary search, we describe our Global Pseudo-task Simulation method in Algorithm 3.

**Algorithm 2** GlobalBS

**Input:** Number of tasks $T$; Task distributions $P_{1:i}$; Local updating method $g$; Current model parameters $\theta_i$ and current memory $\mathcal{M}_{i-1}$; Search stride $\epsilon$.
Synthesize pseudo-tasks from $P_i$ with task distributions.
$start \leftarrow 0$
$end \leftarrow |\mathcal{M}|/i$
Set bounds for the stride: $\epsilon \leftarrow \max(min\_\epsilon, \min(max\_\epsilon, \epsilon))$
Accuracy dictionary: $acc\_dict \leftarrow \varnothing$
**while** $end - start \geq \epsilon$ **do**
    $next \leftarrow (start + end)/2$
    **if** $next$ not in $acc\_dict$ **then**
        $loss, acc \leftarrow \text{GlobalSim}(next, \tilde{P}_{(i+1):T}, g, \theta_i, \mathcal{M}_{i-1})$
        $acc\_dict \leftarrow acc\_dict \cup \{next : acc\}$
    **else**
        $acc \leftarrow acc\_dict[next]$
    **end if**
    **if** $next - \epsilon$ not in $acc\_dict$ **then**
        $left\_loss, left\_acc \leftarrow \text{GlobalSim}(next - \epsilon, \tilde{P}_{(i+1):T}, g, \theta_i, \mathcal{M}_{i-1})$
        $acc\_dict \leftarrow acc\_dict \cup \{next - \epsilon : left\_acc\}$
    **else**
        $left\_acc \leftarrow acc\_dict[next - \epsilon]$
    **end if**
    **if** $next + \epsilon$ not in $acc\_dict$ **then**
        $right\_loss, right\_acc \leftarrow \text{GlobalSim}(next + \epsilon, \tilde{P}_{(i+1):T}, g, \theta_i, \mathcal{M}_{i-1})$
        $acc\_dict \leftarrow acc\_dict \cup \{next + \epsilon : right\_acc\}$
    **else**
        $right\_acc \leftarrow acc\_dict[next + \epsilon]$
    **end if**
    **if** $left\_acc < acc$ **then**
        $end \leftarrow next$
        **continue**
    **else if** $right\_acc < acc$ **then**
        $start \leftarrow next$
        **continue**
    **else**
        **break**
    **end if**
**end while**
$s_i \leftarrow argmin_{acc}(acc\_dict)$
**return** Switching point: $s_i$

---

**Algorithm 3** Global Pseudo-task Simulation (GPS)

**Input:** Number of tasks $T$; Task distributions $P_i, i \in \mathcal{T}$; Local updating method $g(\cdot)$.
Initialize parameters $\theta_0$
Initialize memory $\mathcal{M}_0 = \varnothing$
$i \leftarrow 1$
**while** $i \leq T$ **do**
    Local update: $\theta_i \leftarrow g(\theta_{i-1}, P_i, \mathcal{M}_{i-1})$
    Find switching point: $s_i \leftarrow \text{GlobalBS}(T, P_{1:i}, g, \theta_i, \mathcal{M}_{i-1}, |\mathcal{M}|/5i)$
    Build memory: $\mathcal{M}_i \leftarrow \text{BuildM}(\mathcal{M}_{i-1}, s_i)$
    $i \leftarrow i + 1$
**end while**
**return** Model parameters: $\theta_T$

# B  Validation of the Switching Point Existence

As a validation of our switching point existence, we further show the switching point of other benchmarks in our experiments. For each benchmark, we plot the global loss $L^G$ as a function of $a_i$ and select 5 different tasks $t_i$. Each plot shows clearly the switching point in Fig. 4.

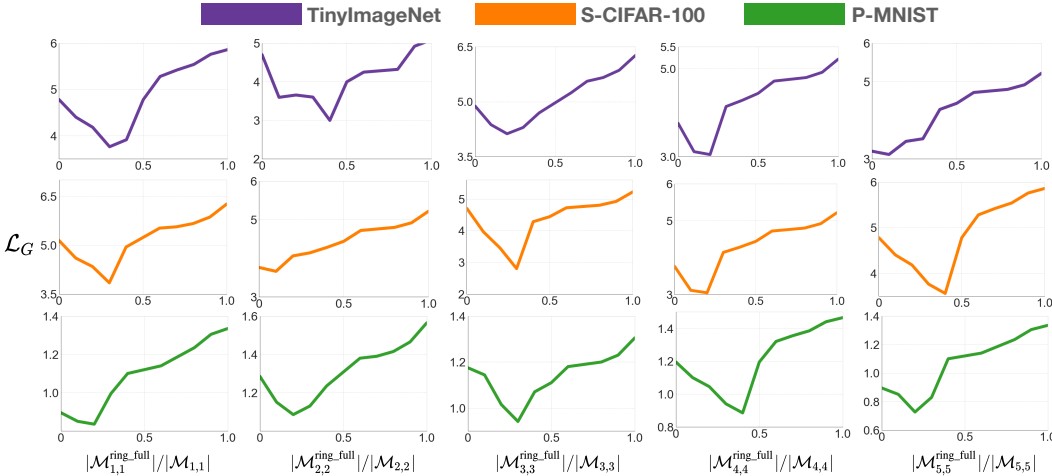

Figure 4: Switching points of the first 5 tasks in three evaluation benchmarks: TinyImageNet, S-CIFAR-100, P-MNIST. We show the change of the global loss, $\mathcal{L}^G$ w.r.t. the ratio of ER-Ring-Full in the memory.

# C  Validation of the Simulation Method

In this section, we provide the empirical supporting evidences for our hypotheses of the simulation method.

## C.1  Task Difficulties

First, we show the task difficulties in the evaluated benchmarks have small variations, as in Table 3. For P-MNIST, S-CIFAR-100 and TinyImageNet, we evaluate the first 5 tasks end-to-end for simplicity. For S-CIFAR-10, we evaluate all the tasks end-to-end. Further, we evaluate the difficulty along the

Table 3: Accuracy and variance of accuracy of tasks from four vision benchmarks trained end-to-end.

| Dataset | Task 1 | Task 2 | Task 3 | Task 4 | Task 5 | Variance |
|---|---|---|---|---|---|---|
| **P-MNIST** | 97.48 | 97.28 | 97.33 | 97.78 | 97.53 | 0.03 |
| **S-CIFAR-10** | 98.20 | 94.85 | 96.50 | 98.90 | 98.15 | 2.18 |
| **S-CIFAR-100** | 85.70 | 87.70 | 88.10 | 88.10 | 86.20 | 1.02 |
| **TinyImageNet** | 78.20 | 76.80 | 77.30 | 76.50 | 77.20 | 0.33 |

pseudo-task sequence synthesized from the first task of S-CIFAR-100 by permutation, rotation and blurring, we generate 5 tasks for each simulation method. The results in Table 4 shows permutation and rotation generate tasks with similar difficulties as the original S-CIFAR-100 tasks, while blurring generate tasks with increasing difficulties as the task sequence grows.

## C.2  Zero-shot Transfer Ability of Tasks

Next, we explore the zero-shot transfer ability of the pseudo-tasks *vs.* the real tasks. We evaluate the model trained on the first two tasks of S-CIFAR-100 on task sequences different by different simulation methods.

Table 4: Accuracy and variance of accuracy of pseudo-tasks synthesized by different methods from the first task of S-CIFAR-100. The Task $i$ column refers to the end-to-end training of the pseudo-task $i$.

| Method | Task 1 | Task 2 | Task 3 | Task 4 | Task 5 | Variance |
|--------|--------|--------|--------|--------|--------|----------|
| Permutation | 85.50 | 85.70 | 85.30 | 86.10 | 85.10 | 0.11 |
| Rotation | 85.40 | 85.40 | 85.70 | 84.90 | 84.50 | 0.18 |
| Blurring | 83.90 | 81.70 | 78.90 | 74.40 | 69.50 | 26.80 |

Table 5 shows that the permutation pseudo-tasks and the real tasks both allows zero transfer ability, as the random guess accuracy of a 10-class classification is 10%. Rotation, instead, creates a task sequence that allows nearly perfect zero-shot transfer ability. Blurring creates a task sequence which allows some zero-shot transfer ability from the beginning, but it gradually reduces to a random guess as task difficulty grows.

Table 5: Zero-shot transfer ability of different simulation methods after training task $t_1$ and task $t_2$ on S-CIFAR-100. Numbers are the accuracy of the task.

| Dataset | Method | Task 3 | Task 4 | Task 5 | Task 6 | Task 7 |
|---------|--------|--------|--------|--------|--------|--------|
| | Real | 11.80 | 10.40 | 10.30 | 9.40 | 9.70 |
| S-CIFAR-100 | Permutation | 10.50 | 10.40 | 10.30 | 10.10 | 11.10 |
| | Rotation | 75.40 | 78.10 | 77.70 | 79.90 | 80.50 |
| | Blurring | 70.90 | 65.70 | 52.90 | 30.40 | 15.50 |

# D    Additional Experimental Results

## D.1    Accuracy of GPS with Small Memory Buffer

We also conduct experiments of GPS to evaluate its performance when the memory buffer is relatively small, as shown in Table 6. With a small size memory buffer, GPS does not show significant improvement. One reason is the switching point $s_j$ is very close to 1, *i.e.*, taking the pure ER-Ring-Full policy is good enough. Another cause is our base policy assumption does not work very well under the small memory buffer size as model becomes more sensitive to points selection under small $\mathcal{M}$.

Table 6: Accuracy of GPS using permutation and ER baselines on four datasets with smaller buffer sizes.

| Method $\mid\mathcal{M}\mid$ | P-MNIST 100 | S-CIFAR-10 20 | S-CIFAR-100 200 | TinyImageNet 200 |
|--------|--------|--------|--------|--------|
| ER-Res | $65.59_{\pm 1.38}$ | $80.68_{\pm 2.28}$ | $64.99_{\pm 1.74}$ | $38.60_{\pm 0.74}$ |
| ER-Ring-Full | $66.10_{\pm 1.36}$ | $81.30_{\pm 1.98}$ | $65.95_{\pm 0.96}$ | $\mathbf{39.85_{\pm 0.78}}$ |
| ER-Hybrid | $66.30_{\pm 1.21}$ | $\mathbf{81.43_{\pm 2.54}}$ | $66.30_{\pm 1.32}$ | $39.75_{\pm 0.54}$ |
| **GPS** | $\mathbf{66.31_{\pm 1.11}}$ | $81.35_{\pm 1.56}$ | $\mathbf{66.51_{\pm 0.88}}$ | $39.83_{\pm 0.45}$ |

## D.2    Results of GPS with DER++, HAL and Baselines

We put complete results of GPS+DER, GPS+DER++, GPS+HAL and baselines in Table 7. Note the A-GEM [10], iCaRL [37] and GSS [3] use the same memory size as the ER series for fair comparison. The results stand as a complete empirical support to illustrate that the performance of other ER variants have been improved after using our GPS method.

Table 7: Accuracy of GPS using permutation incorporating DER, DER++ [6] and HAL [9], comparing to other methods. '-' indicates experiments we were unable to run, due to compatibility issues (e.g. Domain IL for iCaRL) or intractable training time or memory utilization (e.g. OGD, GSS on TinyImageNet).

| | P-MNIST | S-CIFAR-10 | S-CIFAR-100 | TinyImageNet |
|---|---|---|---|---|
| oEWC | $69.21_{\pm 2.92}$ | $62.97_{\pm 3.55}$ | $55.37_{\pm 2.71}$ | $20.81_{\pm 0.95}$ |
| iCaRL | - | $88.97_{\pm 2.77}$ | $78.21_{\pm 1.01}$ | $38.77_{\pm 3.68}$ |
| GSS | $86.34_{\pm 4.28}$ | $87.80_{\pm 2.71}$ | $77.34_{\pm 3.21}$ | - |
| A-GEM | $77.36_{\pm 1.28}$ | $83.87_{\pm 1.55}$ | $69.61_{\pm 1.47}$ | $25.30_{\pm 0.87}$ |
| OGD | $81.52_{\pm 2.21}$ | - | - | - |
| | **P-MNIST** | **S-CIFAR-10** | **S-CIFAR-100** | **TinyImageNet** |
| $|\mathcal{M}|$ | 1000 | 200 | 2000 | 2000 |
| HAL | $87.69_{\pm 1.34}$ | $82.51_{\pm 3.20}$ | - | - |
| **GPS+HAL** | $\mathbf{88.23_{\pm 1.03}}$ | $\mathbf{82.77_{\pm 0.93}}$ | - | - |
| DER | $90.47_{\pm 0.69}$ | $91.04_{\pm 0.18}$ | $81.78_{\pm 0.50}$ | $60.40_{\pm 1.08}$ |
| **GPS+DER** | $\mathbf{90.27_{\pm 0.78}}$ | $\mathbf{91.33_{\pm 0.13}}$ | $\mathbf{83.39_{\pm 0.44}}$ | $\mathbf{60.89_{\pm 1.06}}$ |
| DER++ | $91.14_{\pm 0.22}$ | $92.06_{\pm 0.20}$ | $82.20_{\pm 0.89}$ | $60.67_{\pm 1.08}$ |
| **GPS+DER++** | $\mathbf{91.64_{\pm 0.16}}$ | $\mathbf{92.37_{\pm 0.10}}$ | $\mathbf{83.53_{\pm 0.64}}$ | $\mathbf{61.01_{\pm 0.98}}$ |

# E   New Base Policies based on Curriculum Learning

To further test the power of GPS, we substitute the base policies with two novel memory construction methods designed by us based on curriculum learning [16], ER-CurRes and ER-CurRing-Full. The inspiration of these two methods comes from recent findings that curriculum can help when noisy data are present [50, 31, 41]. We believe data points of future task can be viewed as noisy interference for samples stored in $\mathcal{M}$. In these two policies, curricular easy points of each task are picked as candidates for $\mathcal{M}$.

## E.1   Algorithm

**Curricular Easy Samples**   We rank data examples from easy to hard based on the implicit curricula [50]. Specifically, we first record the learned epochs as an attribute of an example, which is the earliest epoch in training where a model correctly predicts this example for that and subsequent epochs till now. As the learned epoch is a positive integer attribute, it is defined as a subset of the totally ordered set $\mathbb{Z}^+$. We also record the current loss of each example as another attribute. The loss attribute is defined as a subset of the totally ordered set $\mathbb{R}$. The ranking of examples is based on the lexicographical order on the Cartesian product of the two attributes, *i.e.*, first sorting the examples by the learned epoch attribute, and then ordering examples within the same ranked epoch by their losses. As a result, each training example of a task would be associated with an unique ranking. Also, the ranking would be updated after each epoch.

**ER-CurRes**   The ER-CurRes algorithm is shown in Algorithm. 4. Different from ER-Res which stores examples sampled from the whole distribution $P_i$ of a task $t_i$ [11], we sample subset of data points from an "easy pool", *i.e.* $P_i^{\text{easy}}$. The size is calculated by the dataset size $|\mathcal{D}_i|$ multiplying a hyperparameter $\gamma$, whose value is reported for each evaluation dataset in Section F. Compared to taking the top few easiest points of each class, sampling from the pool utilizes the benefits of randomness [4, 50]. Suppose we train a total of $k$ epochs of task $t_i$, in order to obtain a smooth transition and the samples based on a more stable curriculum ranking, we take the examples obeying ER-Res policy (*i.e.*, samples from $P_i$) for the first $\lceil k/2 \rceil$ epochs, and take the examples obeying ER-CurRes policy (*i.e.*, samples from $P_i^{\text{easy}}$) for the last $\lfloor k/2 \rfloor$ epochs. When we finish training, we replace the examples of task $t_i$ in the memory which are not from $P_i^{\text{easy}}$.

**ER-CurRing-Full**   Likewise, as shown in Algorithm. 5, ER-CurRing-Full follows the ER-Ring-Full strategy in the first $\lceil k/2 \rceil$ epochs to fill a FIFO memory [11]. After that, we substitute the memory slots with points from the easy pool of each observed class. In the construction of the easy pool,

---

**Algorithm 4** ER-CurRes (for a single task)

---

**Input:** Reservoir memory buffer $\mathcal{M}$; Number of epochs $k$; Task distribution $P$; Dataset size $|\mathcal{D}|$; Batch size $B$; Portion of easy data $\gamma$; Model parameters $\theta$; Seen examples $N$.

Initialize a random easy pool $P^{\text{easy}}$ from $P$

**for** $ep \in \{1, ..., k\}$ **do**
    **for** $iter \in 1, ..., |\mathcal{D}|/B$ **do**
        Sample a batch $B_P$ from $P$ and a batch $B_{\mathcal{M}}$ from $\mathcal{M}$
        Update $\theta$ with $B_P \cup B_{\mathcal{M}}$
        **if** $ep \leq \lceil k/2 \rceil$ **then**
            Update $\mathcal{M}$ with a probability $|\mathcal{M}|/N$ for each examples in $B_P$
            $N = N + 1$
        **else**
            Update $\mathcal{M}$ with a probability $|\mathcal{M}|/(\gamma * N)$ for each examples in $B_P \cap P^{\text{easy}}$
            $N = N + 1/\gamma$
        **end if**
    **end for**
    Update $P^{\text{easy}}$: order examples based on the implicit curriculum and select the first $\gamma|\mathcal{D}|$
**end for**
Select the memory for the current task as $\mathcal{M}_{\text{now}}$ and the memory for all previous tasks as $\mathcal{M}_{\text{past}}$
**for** $idx \in \mathcal{M}_{\text{now}}$ **do**
    **if** $idx \notin P^{\text{easy}}$ **then**
        Replace the slot in $\mathcal{M}_{\text{now}}$ with samples from $(P^{\text{easy}} - \mathcal{M}_{\text{now}})$
    **end if**
**end for**
**Return** Updated $\theta$ and $\mathcal{M} = \mathcal{M}_{\text{now}} \cup \mathcal{M}_{\text{past}}$

---

instead of taking the easiest $\gamma|\mathcal{D}|$ examples as in ER-CurRes, we use the easiest $\gamma|\mathcal{D}|/C$ examples of each class based on the ranking, where $C$ is the number of classes.

### E.2 Experimental Results of GPS w/ Cur

The accuracy comparison between GPS w/ Cur and GPS w/ ER-CurRes, ER-CurRing-Full are shown in Table 8. From the table, we can see GPS w/ Cur outperforms both ER-CurRes and ER-CurRing-Full in both datasets. Besides, GPS with the blending of curriculum policies do not significantly outperform the blending of ER-Ring-Full and ER-Res. It implies that applying ER-Ring-Full and ER-Res as base policies are probably enough for the current benchmarks under the task- and domain-IL setups.

Table 8: Accuracy of GPS using curriculum-based policies *vs.* the corresponding baselines.

| | **P-MNIST** | **S-CIFAR10** | **S-CIFAR100** | **TinyImageNet** |
|---|---|---|---|---|
| $|\mathcal{M}|$ | 1000 | 200 | 2000 | 2000 |
| ER-CurRes | $86.78_{\pm 0.49}$ | $91.47_{\pm 0.20}$ | $81.38_{\pm 0.51}$ | $58.89_{\pm 0.03}$ |
| ER-CurRing-Full | $86.16_{\pm 0.49}$ | $91.70_{\pm 0.50}$ | $81.16_{\pm 0.65}$ | $58.03_{\pm 0.42}$ |
| **GPS w/ Cur** | $\mathbf{87.35_{\pm 0.18}}$ | $\mathbf{92.08_{\pm 0.17}}$ | $\mathbf{82.34_{\pm 0.79}}$ | $\mathbf{59.88_{\pm 0.03}}$ |
| | **P-MNIST** | **S-CIFAR10** | **S-CIFAR100** | **TinyImageNet** |
| $|\mathcal{M}|$ | 100 | 20 | 200 | 200 |
| ER-CurRes | $65.34_{\pm 0.69}$ | $81.59_{\pm 3.23}$ | $65.43_{\pm 1.37}$ | $38.75_{\pm 0.98}$ |
| ER-CurRing-Full | $66.42_{\pm 1.03}$ | $81.54_{\pm 2.46}$ | $66.73_{\pm 0.12}$ | $39.54_{\pm 0.57}$ |
| **GPS w/ Cur** | $\mathbf{66.52_{\pm 0.54}}$ | $\mathbf{81.68_{\pm 1.98}}$ | $\mathbf{67.08_{\pm 0.36}}$ | $\mathbf{39.80_{\pm 0.49}}$ |

---

**Algorithm 5** ER-CurRing-Full (for a single task)

---

**Input:** Ring-Full memory buffer $\mathcal{M}$; Number of epochs $k$; Task distribution $P$; Dataset size $|\mathcal{D}|$; Batch size $B$; Portion of easy data $\gamma$; Model parameters $\theta$.

Initialize a random easy pool $P^{\text{easy}}$ from $P$

Reallocate the memory $\mathcal{M}_{\text{past}}$ for all previous tasks, and allocate the memory $\mathcal{M}_{\text{now}}$ for the current task

**for** $e \in \{1, ..., k\}$ **do**

    **for** $iter \in 1, ..., |\mathcal{D}|/B$ **do**

        Sample a batch $B_P$ from $P$ and a batch $B_{\mathcal{M}}$ from $\mathcal{M}$

        Update $\theta$ with $B_P \cup B_{\mathcal{M}}$

        **if** $e \leq \lceil k/2 \rceil$ **then**

            Update $\mathcal{M}_{\text{now}}$ with $B_P$

        **else**

            Update $\mathcal{M}_{\text{now}}$ with $B_P \cap P^{\text{easy}}$

        **end if**

    **end for**Update $P^{\text{easy}}$: order examples based on the implicit curriculum and select the first $\gamma$ portion of each class

**end for**

**for** $idx \in \mathcal{M}_{\text{now}}$ **do**

    **if** $idx \notin P^{\text{easy}}$ **then**

        Replace the slot in $\mathcal{M}_{\text{now}}$ with samples from $(P^{\text{easy}} - \mathcal{M}_{\text{now}})$

    **end if**

**end for**

**Return** Updated $\theta$ and $\mathcal{M} = \mathcal{M}_{\text{now}} \cup \mathcal{M}_{\text{past}}$

---

## F Experimental Details

### F.1 Simulation Details

In experiments, we set the number of examples in each synthesized pseudo-task the same as the size of the memory buffer, *i.e.*, if $|\mathcal{M}| = 1000$, we then generate 1000 examples for each pseudo-task. For computational efficiency, we set the number of training epochs small in the simulated training process. We train 1 epoch for pseudo-tasks synthesized in the P-MNIST dataset, 5 epochs for pseudo-tasks in S-CIFAR-10, S-CIFAR-100 and TinyImageNet. As for the batch size, the optimizer and the learning step during the simulation process, they are all the same as in the real training process.

### F.2 Other Hyperparameters

We disclose the experimental hyperparameters values not reported in the main manuscript in Table 9. In the table, $\gamma$ in the 'Cur-' series methods is the easy pool ratio of the curriculum-based policies as we discussed in Section E, while other symbols refer to the respective methods. In all the experimental evaluation by accuracy, reported numbers are averaged over 5 runs.

### F.3 Time Measurement

We measure our training and simulation time for each dataset in a single NVIDIA Tesla K80 GPU for fair comparison. The time we report is the total processing time averaged on 5 runs, assessed in wall-clock time (seconds) at the end of the last task and then converted into minutes.

## G ER-Res & ER-Ring-Full Algorithms

We put the algorithms of ER-Res [11] and ER-Ring-Full [11] for a single task in Algorithm 6 and Algorithm 7 for reference.

Table 9: Other hyperparameters used in our experiments.

| Dataset | Method | Parameters |
|---|---|---|
| **P-MNIST** | CurER-Res | $\gamma$: 0.2 |
| | CurER-Ring-Full | $\gamma$: 0.1 |
| | GPS w/ Cur | $\gamma$: 0.2 |
| | DER | $\alpha$: 0.5 |
| | DER++ | $\alpha$: 1.0 $\beta$: 0.5 |
| | GPS+DER | $\alpha$: 0.5 |
| | GPS+DER++ | $\alpha$: 1.0 $\beta$: 0.5 |
| | HAL | $\lambda$: 0.1 $\beta$: 0.5 $\gamma$: 0.1 |
| | GPS+HAL | $\lambda$: 0.1 $\beta$: 0.5 $\gamma$: 0.1 |
| | oEWC | $\lambda$: 0.7 $\gamma$: 1.0 |
| | GSS | $gmbs$: 10 $nb$: 1 |
| | OGD | stored gradients : 100/task (perm) |
| **S-CIFAR-10** | CurER-Res | $\gamma$: 0.2 |
| | CurER-Ring-Full | $\gamma$: 0.1 |
| | GPS w/ Cur | $\gamma$: 0.2 |
| | DER | $\alpha$: 0.3 |
| | DER++ | $\alpha$: 0.1 $\beta$: 0.5 |
| | GPS+DER | $\alpha$: 0.3 |
| | GPS+DER++ | $\alpha$: 0.1 $\beta$: 0.5 |
| | HAL | $\lambda$: 0.1 $\beta$: 0.5 $\gamma$: 0.1 |
| | GPS+HAL | $\lambda$: 0.1 $\beta$: 0.5 $\gamma$: 0.1 |
| | oEWC | $\lambda$: 0.7 $\gamma$: 1.0 |
| | iCaRL | $wd$: 0 |
| | GSS | $gmbs$: 32 $nb$: 1 |
| **S-CIFAR-100** | CurER-Res | $\gamma$: 0.2 |
| | CurER-Ring-Full | $\gamma$: 0.1 |
| | GPS w/ Cur | $\gamma$: 0.2 |
| | DER | $\alpha$: 0.5 |
| | DER++ | $\alpha$: 0.5 $\beta$: 0.5 |
| | GPS+DER | $\alpha$: 0.5 |
| | GPS+DER++ | $\alpha$: 0.5 $\beta$: 0.5 |
| | HAL | $\lambda$: 0.1 $\beta$: 0.5 $\gamma$: 0.1 |
| | GPS+HAL | $\lambda$: 0.1 $\beta$: 0.5 $\gamma$: 0.1 |
| | oEWC | $\lambda$: 0.7 $\gamma$: 1.0 |
| | iCaRL | $wd$: $10^{-5}$ |
| | GSS | $gmbs$: 32 $nb$: 1 |
| **TinyImageNet** | CurER-Res | $\gamma$: 0.2 |
| | GPS w/ Cur | $\gamma$: 0.2 |
| | CurER-Ring-Full | $\gamma$: 0.1 |
| | DER | $\alpha$: 0.1 |
| | DER++ | $\alpha$: 0.1 $\beta$: 0.5 |
| | GPS+DER | $\alpha$: 0.1 |
| | GPS+DER++ | $\alpha$: 0.1 $\beta$: 0.5 |
| | HAL | $\lambda$: 0.1 $\beta$: 0.5 $\gamma$: 0.1 |
| | GPS+HAL | $\lambda$: 0.1 $\beta$: 0.5 $\gamma$: 0.1 |
| | oEWC | $\lambda$: 0.7 $\gamma$: 1.0 |
| | iCaRL | $wd$: $10^{-5}$ |

**Algorithm 6** ER-Res. $|\mathcal{M}|$ is the number of examples the memory can store, $t$ is the task id, $n$ is the number of examples observed so far in the data stream, and $B$ is the input mini-batch.

---

1: **procedure** UPDATEMEMORY($|\mathcal{M}|, t, n, B$)
2:  $j \leftarrow 0$
3:  **for** $(x, y)$ in $B$ **do**
4:    $M \leftarrow \mathcal{M}$
5:   **if** $M < |\mathcal{M}|$ **then**
6:     $\mathcal{M}$.append($x, y, t$)
7:   **else**
8:     $i = randint(0, n + j)$
9:    **if** $i < |\mathcal{M}|$ **then**
10:      $\mathcal{M}[i] \leftarrow (x, y, t)$
11:    **end if**
12:   **end if**
13:   $j \leftarrow j + 1$
14:  **end for**
15:  **return** $\mathcal{M}$
16: **end procedure**

---

**Algorithm 7** ER-Ring-Full.

---

1: **procedure** UPDATEMEMORY($|\mathcal{M}|, t, n, B$)
2:  **for** $(x, y)$ in $B$ **do**
3:   Divide memory into FIFO stacks $\mathcal{M}[y]$, where $|\mathcal{M}[y]| = |\mathcal{M}|/t$
4:   $\mathcal{M}[y]$.append($x$)
5:  **end for**
6:  **return** $\mathcal{M}$
7: **end procedure**

---